# Youth Perspectives on Collaborative Consumption: A Study on the Attitudes and Behaviors of the Romanian Generation Z

**Daniel Bulin** [1,2], **Georgică Gheorghe** [1,*] , **Adrian Lucian Kanovici** [1], **Adrian Bogdan Curteanu** [3], **Oana-Diana Curteanu** [1] and **Robert-Ionuţ Dobre** [2]

1   Faculty of Business and Tourism, Bucharest University of Economic Studies, 010374 Bucharest, Romania; daniel.bulin@com.ase.ro (D.B.); adrian.kanovici@rei.ase.ro (A.L.K.); oana.crismariu@com.ase.ro (O.-D.C.)
2   Institute for World Economy, Romanian Academy, 010071 Bucharest, Romania; robert_dobre@yahoo.com
3   Faculty of Theoretical & Applied Economics, Bucharest University of Economic Studies, 010374 Bucharest, Romania; curteanuadrian11@stud.ase.ro
*   Correspondence: georgica.gheorghe@com.ase.ro; Tel.: +40-753-096-004

**Abstract:** With the emergence of the sharing economy, a significant change in consumer behavior can be observed worldwide, which has a considerable impact on various industries. The rise of the sharing economy has changed the way people experience transport services, with ridesharing being a catalyst for change. In Romania, the debut of Uber in 2015 sparked controversy and led to legal regulations that were adapted to local specificities, highlighting the adaptability of ridesharing platforms to different legal frameworks. In the context of this development, the views and perceptions of Generation Z will be crucial in determining the direction in which this conflict between disruptive models and traditional players in the transport sector develops. The article deals with business models based on collaborative consumption, with a focus on ridesharing, and examines the attitudes, perceptions, and behavior of Romanian youths (aged 18–26) towards these models. The aim of the study is to determine the opinion of young Romanians on collaborative consumption in transport services—ridesharing (Uber case)—and their attitude towards the ethical controversies related to Uber's business model. A quantitative research approach was chosen, and an exploratory study was conducted using a questionnaire, with the non-probabilistic sample consisting of relevant observation units aged 18–26 years. The results show that almost 90% of the young Romanians surveyed use Uber and are satisfied with the quality, convenience, and speed of the service. Despite the positive attitude, there is a paradoxical tendency among respondents to regulate ridesharing services in a similar way to traditional taxis. Ethical considerations show that respondents tend to neutralize perceptions and justify the emergence of new models as normal and beneficial for competition and consumers.

**Keywords:** UBER; collaborative consumption; youth; quantitative research; ridesharing





## 1. Introduction

The sharing economy is having a major impact on many industries and businesses around the world, and as a result there is a growing interest in business models for collaborative consumption. In particular, understanding the factors that influence consumer behaviors, attitudes and intentions towards collaborative consumption has become an important research objective.

The rise of the sharing economy has revolutionized the way people around the world interact with different services. The emergence of collaborative consumption models such as ridesharing has heralded a new era in transport [1]. It is important to understand the factors that influence the behavior of Generation Z in this paradigm. Their dynamic nature, born in an age characterized by connectivity but also concern for sustainability, makes them an important subject for exploring their attitudes towards ridesharing services. As we delve into the nuances of collaborative consumption, it becomes clear that the collaborative economy is more than a trend—it is a transformative force impacting societies worldwide.

According to [2] the proportion of European internet users using collaborative economy platforms increased from 22.7% in 2020 to 26.4% in 2022. With the increasing focus on sustainable and community-oriented practices, exploring Generation Z's perception and behavior towards ridesharing models becomes a captivating research subject.

The emergence of Uber and other ridesharing services has sparked much controversy in traditional transport markets, including the taxi industry: difficulties in complying with traditional taxi laws, accusations of unfair competition as Uber operates in a more flexible environment and is not subject to the same strict regulations and taxes as traditional taxi services, controversy over the status of drivers as self-employed, concerns over the driver verification process and passenger safety on the platform, and the impact of ridesharing services on urban transport and infrastructure.

Uber was launched in Romania in 2015 and has met with various reactions and controversies over the years. Although traditional taxi drivers protested against the service, considering it unfair competition, Uber managed to expand its network to 22 major cities in Romania by 2024, with Uber users covering a distance of over 826 million kilometers on the country's roads [3]. Subsequently, there were heated discussions about the legality of Uber services in Romania, which eventually led to massive protests and changes to road traffic laws in order to regulate ridesharing services more clearly and bring them under control [4]. Uber's experience in Romania mirrors global trends where ridesharing platforms have had to adapt to the local legal and regulatory environment, in some cases leading to tensions with traditional taxi services [5,6].

As the sharing economy changes the way we experience transport services, the importance of young people's opinions and perceptions in the context of the clash between disruptive ride-sharing models and traditional players such as taxis becomes clear. Young people can play an important role in determining the direction of this development. Their views on ethical issues and social responsibility can have a major impact on the success and acceptance of these innovative services.

In the competition between traditional players, such as taxis, and ridesharing providers, the perspective of young people is becoming a key factor. How they perceive equity, community benefit and the role of each type of service in their social and urban environment will determine the evolution of this conflict. Young people's opinions will therefore not only be an indicator of consumer preferences, but also a crucial factor in resolving the ethical and competitive tensions between the old and new transport models.

Young people are an interesting topic for a geographically located study of the Romanian market, especially due to the rapid expansion of the collaborative economy, particularly ridesharing services in major cities, especially in the capital. Analyzing the attitudes and perceptions of this demographic segment within the collaborative economy reveals a complex picture and thus makes an important contribution to understanding the global dynamics of this industry.

The aim of this article is to explore the attitudes, perceptions, and behaviors of Romanian Gen Z towards ridesharing business models based on the sharing economy and collaborative consumption.

Specific objectives:

- Identifying the opinion of young Romanians on transport services based on business models of collaborative consumption (ridesharing);
- Determining the opinion and position of young Romanians in relation to the ethical controversies of Uber business models.

Paper structure: This introductory section is followed by an overview of the specialized literature, which serves as a basis for the research design and the development of the questionnaire. The methodology is then presented, outlining the research questions, the statistical hypotheses, the research design, the data collection process, and the research variables. The results obtained are then highlighted in a detailed section and discussed in specific subsections. These subsections include the presentation of the research sample, the use and perception of Uber services, and conflicts and ethical controversies related to Uber

taxis. Conclusions are drawn in close relation to the research hypotheses and finally the implications for management and business as well as the limitations of the research are emphasized.

## 2. Literature Review

Several studies have investigated the reactions of Millennials and Generation Z to the sharing economy phenomenon, and the results have been quite contradictory. Ref. [7] introduced the concept of Collaborative Consumption as being those events in which one or more persons consume economic goods or services in the process of engaging in joint activities with one or more others. Ref. [8] define Collaborative Consumption as the rapid explosion in swapping, sharing, bartering, trading and renting being reinvented through the latest technologies and peer-to-peer marketplaces in ways and on a scale never before possible emphasizing the significance of trust, convenience, and sustainability in driving participation and organizing the thousands of examples of Collaborative Consumption from around the world into three systems—product service systems, redistribution markets and collaborative lifestyles.

A 2017 study by [9], which analyzed the opinions and interaction with the sharing economy of 850 consumers in the UK and the US, revealed a paradox: although millennials are considered the "core" customers" of collaborative consumption who use the platforms based on the new business models of the sharing economy, their opinions do not seem to be taken into account in the plans of those developing the platforms: the process of identity verification or the lack of speed of transactions (related to the impatience of young people) were mentioned as problems that cause them to refrain from transactions.

Ref. [10] conducted a study focusing on millennials' perspectives on collaborative consumption and examined how values and attitudes relate to their behavioral intentions. The results show that millennials' perceptions of collaborative consumption significantly influence their attitudes and empathy towards these new business models. Their research illustrates a direct and positive relationship between intention, attitude, and empathy towards collaborative consumption. Interestingly, no relationship was found between empathy and attitude. The study used six measures including utility, hedonic value, symbolic value, attitude, empathy, purchase intention, and a control variable (familiarity) to analyze these relationships.

While ref. [11] suggests that support for collaborative consumption platforms is driven by perceived benefits such as free competition and innovation, ref. [12] notes that materialistic consumers, though initially skeptical, may still be willing to explore such models.

Ref. [13] have tested a model for consumers and providers that assumes that five factors—economic benefit, sustainability, pleasure, social relationships and attitude—are the most important prerequisites for the intention to use from the perspective of both consumers and providers. According to the model of [13], the network effect is the only factor that is important from both perspectives—consumer and provider. Sustainability, on the other hand, is only significant in the supplier model and not in the consumer model. These findings have important business implications for companies that want to be successful in the collaborative economy market. They need to consider the social impact of their business (for example, Airbnb customers use the platform solely for personal benefit).

Building on previous research into the relationship between millennials and collaborative consumption business models, our study aimed to identify the attitudes and perceptions of Generation Z members towards the use of such services. We sought to consider both the controversies surrounding their use and the conflicts that arise between these new business models and traditional models.

Recognized as digital natives due to their deep connection with technology, Generation Z seamlessly integrate various technological tools and platforms into their daily lives [14]. As the future target audience and potential drivers, this generation's familiarity with technology suggests they will play a significant role in shaping the future of platforms like Uber. Considering the economic benefits, such as cost savings per ride, Generation Z's

attitude towards mobile apps, including Uber, tend to be more positive than negative [15]. However, their level of awareness regarding price differentials between Uber and traditional taxi services may vary [16].

The conflict between Uber and traditional taxi services has been emblematic of the disruptions caused by technology in established industries. Uber entered the transportation market with a disruptive business model that challenged the dominance of the traditional taxi industry [17]. While traditional taxis operate through regulated systems, Uber introduced a peer-to-peer model facilitated by a smartphone app. This fundamental difference sparked tension and conflict between the two sectors in many countries.

In Spain, the taxi sector strongly opposed sharing the passenger transport market on an equal footing with VTCs (Vehicles for Hire), which has led to unresolved tensions and ongoing disputes. The conflict arose due to the different regulations and market conditions for traditional taxis and ridesourcing services such as Uber, Cabify, and Bolt. The Spanish Supreme Court intervened and suspended the restrictions imposed by the government on VTCs. It emphasized the need for liberalization and fair competition in the transportation sector [18,19]. Similarly, in Italy in 2022, the government's attempt to liberalize the ride-sharing industry faced fierce opposition from taxi unions, which led to the removal of the provision from the competition law due to protests, strikes, and organized outbursts [20]. In 2023, in Mexico, taxi driver unions, with 12,000 members, accused Uber of unfair competition [21], while, in France, around 2480 taxi drivers initiated another legal battle against Uber, alleging unfair competition and seeking substantial damages totaling around 455 million euros [22].

In Romania, since 2019, a law on alternative transportation has been enacted, bringing ride-sharing companies into legality. However, at the beginning of 2024, authorized transporters have requested the Government to take measures to eliminate piracy in transportation and to stop the expansion of ride-sharing activities into other areas, such as freight transport, courier services, and international passenger transport, all of which are outside the law [23].

On the other hand, previous studies have shown that the emergence and development of Uber has triggered a number of controversies. The most significant controversies are related to customer safety [24] and handling of data privacy issues, privacy being a major concern for the Generation Z participant's attitude and behavior towards mobile apps [15]. Other controversies related to Uber's operations include the unfairness and inadequacy of legislation, low Uber's contribution to the state budget due to low prices and lack of regulation [25], the benefits for competition, and the significant role Uber plays in urban transportation.

Ref. [12] have proposed a critical approach to materialism (the importance consumers place on material goods in order to achieve their personal life goals) and price consciousness (the importance consumers place on price when making purchasing decisions) as antecedents of collaborative consumption. Their results show that [12] firstly, there is an inverse relationship between materialism and consumer attitudes toward collaborative consumption. Materialistic consumers tend to view collaborative consumption unfavorably. Conversely, there is a positive and significant relationship between materialism and consumers' intention to engage in collaborative consumption. Furthermore, the research highlights a positive correlation between price consciousness and consumer attitudes towards collaborative consumption. Consumers who are price-conscious are more likely to have favorable attitude towards collaborative consumption. Additionally, price consciousness is directly linked to consumers' intention to participate in collaborative consumption activities. This suggests that people who are more price-conscious are also more likely to actively engage in collaborative consumption practices.

Ref. [13] investigated the motivations of consumption and production in the collaborative economy from the perspective of sustainability of business models, based on a number of variables—motivations from the perspective of consumers and providers, from the literature review, applied to the Airbnb platform.

From the consumer's point of view, the motivations are enjoyment, independence through ownership, modern style and social experience [26]; cost savings, familiarity, trust and utility [27]; price sensitivity [28]; price, functional attributes, unique and local authenticity, novelty, travel splurge, and sharing economy ethos [29]; and subjective norms, perceived behavioral control, perceived value, unique experience expectation, familiarity, and eWOM [30].

From the providers' perspective the motivations are enjoyment of sharing, income, product variety, social experience, social influence [26]; income, social interaction, sharing (unused space) [31]; and economic, social, and environmental [32].

According to [33], from a consumer perspective, motivations can be distinguished as intrinsic or extrinsic, ref. [34] adding that enjoyment and sustainability are part of the intrinsic motivational dimension, while economic benefits and reputation belong to the extrinsic motivational dimension.

Ref. [35] emphasizes the importance of five categories of consumption motives for the collaborative economy: economic motives, social motives, hedonic value, reducing risks and responsibilities and environmental benefits. From the providers' perspective, the mentioned authors identify three main motives: economic benefits, entrepreneurial freedom, and social motives.

Also, ref. [36] conducted a study among Uber drivers in Morocco to identify the main motivators that led them to take part in the sharing economy. The key finding was that three motives dominate engagement in collaborative consumption: moral, monetary, and social–hedonic motives.

Ref. [11] starts from the idea that collaborative-based platforms (CBP) are questionable and ethically controversial. From this perspective, the results show that users tend to adopt a "neutralization approach" and use accepted techniques to justify the use of CBP. The authors [11] cite well-known studies [37], adapted from [38], that define the main neutralization techniques related to consumer behavior and attitudes: denial of responsibility, denial of injury, denial of victimization, condemnation of the condemners and appeal to higher loyalties (Table 1).

**Table 1.** Neutralization techniques related to Uber clients' behavior.

| Technique | Literature Review Explanation | Ertz et al. [11] Results |
|---|---|---|
| Denial of responsibility | People are aware that they are not directly responsible for their actions if they cannot control the factors, and therefore do not consider themselves fully responsible for their behavior. | The person recognizes the problems caused by the emergence of Uber, but blames the taxi licensing system. The responsibility therefore lies with the current system: it is not Uber's fault, but the government, the taxis, the unions, the bureaucracy and an inadequate legal framework. |
| Denial of injury | Individuals argue that although their behavior is against the rules, it is not serious as long as it does not appear to directly harm anyone. | Those in favor of Uber's activities deny the seriousness of the complaints and relativize the problem by acknowledging that the losses to the state are minimal. |
| Denial of victims | Individuals relativize accusations by saying that their victims deserve what happens to them, as they believe that the situation is the responsibility of the accused person. | In this situation, people believe that taxi drivers deserve what happens to them and that taxi drivers are directly responsible for Uber's success due to the quality of service. |
| Condemnation of condemners | Individuals deflect the moral question onto the accusers by acknowledging that their behavior is similar to theirs. | The technique uses analogies to show that the taxi industry and the government, which make allegations about the illegal nature of Uber's activities, are guilty in different situations. |
| Appeal to higher loyalties | The individual has the feeling that he is breaking the rules in order to achieve a higher goal. | In this case, Uber supporters call for the freedom of choice for consumers, and free and healthy competition, to eliminate the heavily regulated monopolistic system. |

Source: based on [11] research and results.

"Neutralization" refers to the psychological mechanism by which people justify their deviant behavior by temporarily suspending their moral values or beliefs in order to alleviate feelings of guilt or responsibility.

## 3. Methodology

In order to achieve the objectives of this study and to design the research process, we started from the following research questions:

○    What is the ridesharing behavior of Romanian youth as a collaborative consumption model?

○    What is the consumer attitude towards ridesharing as an alternative transport service?

Based on previous results [4,7], three statistical research hypotheses were formulated:

➢    H1. There are no significant differences between respondents who have used the Uber service and those who have not used it in terms of their perceptions and attitudes towards the service;

➢    H2. There are no significant differences between respondents who have used the Uber service and those who have not used it in terms of their attitudes towards the Uber-taxi conflict;

➢    H3. There are no significant differences between respondents who have used the Uber service and those who have not used it in terms of their perceptions towards the controversies caused by ridesharing services.

**Method.** To achieve the objectives of the paper and to test the hypotheses, an exploratory quantitative study was conducted using a questionnaire as an instrument.

The questionnaire was a suitable instrument for our research objectives (to provide a broad and representative picture of opinions, behaviors, and attitudes in relation to the research topic), providing an efficient way (a large number of respondents in a quick and cost-effective way), standardized (uniformity in the data collection process and allows comparison) and confidential (ensuring anonymity and increasing the accuracy of responses) of data collection.

**Research design and data collection**. A quantitative research approach was chosen, based on a survey conducted among the students of the Business and Tourism Faculty of Bucharest University of Economic Studies. In order to ensure a balanced representation of the target population, a contact strategy was used that involved distributing an appropriate number of questionnaires to students based on approximate gender parity according to the data from the Romanian National Institute of Statistics. In order to ensure a balanced distribution of the sample in terms of gender, each student was given to administrate a total of 10 questionnaires, 5 for female and 5 for male respondents. In order to efficiently manage the collected data, the correctly completed questionnaires were registered on the free online platform isondaje.ro, which provided a secure and easy-to-use environment for data entry and subsequent processing. The language of the questionnaire was Romanian.

The research sample was non-probabilistic, and the research unit was young Romanians who were studying or working in Bucharest at the time of data collection (2020) and were between 18 and 26 years old (part of Generation Z).

Questionnaire design. The questionnaire was designed starting from some relevant questions and items from the literature review [4,7] and divided into 4 sections as follows:

●    Section A. Main characteristics: knowledge of concepts, inclination towards the use of technology and collaborative consumption services.

●    Section B. General attitudes towards collaborative consumption: criteria for choice, behavioral actions, consumer behavior and motivations.

●    Section C. Use and perception of Airbnb services.

●    Section D. Use and perception of the Uber service, positioning towards conflicts and addressing ethical controversies.

Questionnaire items. The questionnaire contained various closed, nominal, ordinal and a Likert scale (5-point) to collect data on the behaviors, opinions and characteristics

of the respondents in the study. Both opinion questions (e.g., the level of familiarity with collaborative economy concepts, attitudes, and preferences regarding the use of collaborative economy services) and factual questions (e.g., information such as previous experience of using collaborative economy services, whether or not respondents have used the platforms, and demographic data such as gender, age, education level, occupation and income) were used.

This paper presents the results of the Section 4, focusing on the following questions related to Uber as the first ridesharing company in Bucharest, Romania (in Table 2 we grouped all the variables used according to the fourth categories indicated below):

❖ Consumption ("Have you used Uber alternative transportation services?"; variable: uber, with yes or no answer options)

❖ Attitudes and perceptions of Uber services ("Whether or not you use Uber, please indicate how much you agree with the following statements");

❖ Students position on the uber taxi conflict ("In big Romanian cities (Bucharest, Cluj), as well as in other European cities, Uber services have been the subject of controversy. Anti-Uber arguments also include customer safety issues. To what extent do you agree with the following statements")

❖ The controversies arising from the disruption of the uber taxi transport market ("Please indicate the extent to which you agree or disagree with the following statements regarding the controversies arising from the conflict between Uber and the traditional taxi"); scale measure, from 1—strongly disagree, to 5—strongly agree.

**Table 2.** Research variables.

| Variables | Statements (Questionnaire) |
|---|---|
| Uber_convenient | Uber services are more convenient. |
| Uber_quality | Uber services have better quality. |
| Uber_costs | Uber services reduce costs. |
| Uber_fast | Uber services are faster. |
| Uber_personal_car | Uber allows me to reduce the use of my car. |
| Uber_clients | Uber pays attention to customers. |
| Uber_Experience | Uber offers a superior transportation experience. |
| Uber_conflict_legislation | Uber's business should be based on taxi services regulations and comply with its specific legislation. |
| Uber_conflict_taxes | Uber should pay the same taxes as taxi companies. |
| Uber_conflict_licence | Uber should be licensed as a taxi. |
| Uber_conflict_drivers | Uber drivers should go through the same certification process as taxi drivers. |
| Uber_conflict_tarrifs | Uber tariffs should be regulated locally, similar to taxi services. |
| Uber_conflict_safe | Uber is a safe service. |
| Uber_controversy_law | Private passenger transport legislation is unfair and unadapted to the business model changes. |
| Uber_controversy_budget | Uber company and Uber drivers contribute to the budget by paying taxes. |
| Uber_controversy_quality | The quality of taxi services is extremely low, and companies and taxi drivers are responsible for the success of the alternative service. |
| Uber_controversy_taxi | The taxi industry is like a carter (monopoly) and companies act strictly in their interest and not in that of their customers. |
| Uber_controversy_competition | The emergence of private transport alternatives is favorable to consumers and market development. |
| Uber_controversy_ban | Uber have now an important role in urban transportation and banning it (or other similar companies) is not a solution. |

Source: by authors, based on [4,7].

Based on the literature review, several hypotheses were tested with the results of the survey. The data were then analyzed using SPSS v.26 and Microsoft Excel v.Microsoft 365.

## 4. Results and Discussion

### 4.1. Research Sample

The total number of respondents was 730, of which 721 provided valid responses (Table 3), between 18 and 26 years, with an average age of 21.6. Broken down by gender, 358 men and 363 women responded, almost 50% each, with no difference in average age. Two thirds of respondents (480) had a high school diploma, while a quarter had a bachelor's degree.

**Table 3.** The structure of the respondents (gender, age, and education).

| Gender | Number | % | Last School Graduated | Number | % |
|---|---|---|---|---|---|
| Men | 358 | 49.7% | general school (8 classes) | 8 | 1.1% |
| Women | 363 | 50.3% | high school | 480 | 66.6% |
| Total | 721 | 100.0% | post secondary school | 16 | 2.2% |
| Man Age mean | 21.8 years | | faculty (bachelor) | 170 | 23.6% |
| Woman Age mean | 21.4 years | | master or postgraduate courses | 47 | 6.5% |
| Age mean | 21.6 years | | Total | 721 | 100.0% |

Source: by authors, based on research.

The structure of the respondents according to income and professional status can be seen in Table 4. It shows that over 90% earned less than 3500 lei per month, and almost 80% less than 2500. As far as professional status is concerned, almost 60% of respondents were students at the time (Bachelor's or Master's program), and more than a third were employees.

**Table 4.** The structure of the respondents (income and professional status).

| Income | Number | % | Professional Status | Number | % |
|---|---|---|---|---|---|
| under 1500 lei | 307 | 42.6% | employee | 316 | 37.0% |
| 1500–2500 lei | 246 | 34.1% | Student | 499 | 58.4% |
| 2501–3500 lei | 100 | 13.9% | entrepreneur | 21 | 2.5% |
| 3501–4500 lei | 41 | 5.7% | free-lancer | 8 | 0.9% |
| over 4500 lei | 27 | 3.7% | Total | 854 | 100.0% |
| Total | 721 | 100.0% | | | |

Source: by authors, based on research.

### 4.2. Uber Service Use and Perception

The survey found that 87.4% of young participants used Uber services, a significantly higher rate than those who did not have access to Uber, and the distribution by gender was balanced (Table 5).

Two thirds of respondents who said they use Uber had a high school diploma (421 out of 630), a quarter had a bachelor's degree (152) and only 6% had a master's degree (39). The proportion of those who had a high school diploma or a bachelor's degree was slightly lower among those who had not used the Uber service, but the differences were not really important.

Most users of Uber services belonged to the first two income categories below 2500 lei/month. Their difference to non-Uber users lay in the higher proportion of people with a medium income (approx.15%) and in the almost 50% proportion of those who earned less than 1500 lei.

**Table 5.** Uber service use by gender.

| Uber Use | | | Gender | | |
|---|---|---|---|---|---|
| | | | **Men** | **Women** | **Total** |
| | Yes | Count | 316 | 314 | 630 |
| | | % of Total | 50.2% | 49.8% | 100% |
| | No | Count | 42 | 49 | 91 |
| | | % of Total | 46.2% | 53.8% | 100% |
| Total respondents | | Count | 358 | 363 | 721 |
| | | % of Total | 49.7% | 50.3% | 100% |

Source: by authors, based on research and SPSS output.

The first aspect assessed in relation to Uber's ride-sharing services was respondents' perceptions and attitudes towards the key features and benefits offered by the platform. On a scale of 1—complete disapproval, to 5—complete approval, survey participants indicated that (Table 6):

➢ The quality of Uber services is the most appreciated element: 61% of respondents fully agreed that Uber offers better services than traditional carriers. Full or partial disagreement accounted for only 3% cumulatively.
➢ A total of 80% of respondents fully or partially agreed that the convenience of Uber services and the transport experience were rated as better than with conventional transport services.
➢ Total or partial disagreement did not cumulatively exceed 10%, except for the opinion on the possibility of reducing the use of the personal car, but overall most respondents fully agreed with the statement.

**Table 6.** The mean values for variable Uber service perception and attitude.

| | **Mean** |
|---|---|
| Uber services are more convenient. | 4.321 |
| Uber services have better quality. | 4.438 |
| Uber services reduce costs. | 3.994 |
| Uber services are faster. | 4.165 |
| Uber allows me to reduce the use of my car. | 3.833 |
| Uber pays attention to customers. | 4.29 |
| Uber offers a superior transportation experience. | 4.309 |

Source: by authors, based on research.

Testing the H1 hypothesis (There are no significant differences between respondents who have used the Uber service and those who have not used it in terms of their perceptions and attitudes towards the service) and looking at the results of the Chi-Square test, the following can be observed (Table 7):

■ There were significant differences between the opinions of respondents who had used the Uber service and those who had not, in terms of their perceptions and attitudes towards this service;
■ The relation between the use of the Uber service and the perceptions and attitudes of respondents was inverse, with
  ✛ Medium–low intensity in relation to opinions on the convenience of services, quality of services, cost reduction, speed of services, customer service and superior experience;

✛ Low intensity compared to the opinion that the discounts offered by the Uber service make it possible to reduce the use of a personal car.

**Table 7.** Pearson chi-square test: Uber perception and attitude variables.

| | UBER | |
|---|---|---|
| **Variable** | **Pearson Chi-Square Value** | **Sig. (2-Tailed)** |
| UBER_CONVENIENT | −0.337 | 0.000 |
| UBER_QUALITY | −0.365 | 0.000 |
| UBER_COSTS | −0.288 | 0.000 |
| UBER_FAST | −0.273 | 0.000 |
| UBER_PERSONAL_CAR | −0.160 | 0.000 |
| UBER_CLIENTS | −0.291 | 0.000 |
| UBER_EXPERIENCE | −0.329 | 0.000 |

Source: by authors, based on research and SPSS output.

In terms of the distribution of responses, there were significant differences between the opinions of those who had used Uber services and those who had not, as in the first case, respondents mostly opted for total agreement (5), or a neutral position (3) or, at most, partial agreement (4) (Table 8).

**Table 8.** Uber perception and attitude variable distribution.

| **Variable** | | | **1** | **2** | **3** | **4** | **5** |
|---|---|---|---|---|---|---|---|
| Uber_convenient | yes | % | 1.1 | 2.2 | 10.6 | 23.7 | 62.4 |
| | no | % | 5.5 | 8.8 | 39.6 | 24.2 | 22.0 |
| Uber_quality | yes | % | 0.2 | 1.6 | 7.3 | 24.8 | 66.2 |
| | no | % | 1.1 | 11.0 | 34.1 | 28.6 | 25.3 |
| Uber_costs | yes | % | 1.6 | 6.0 | 18.4 | 27.5 | 46.5 |
| | no | % | 7.7 | 17.6 | 34.1 | 29.7 | 11.0 |
| Uber_fast | yes | % | 1.3 | 3.8 | 13.8 | 29.0 | 52.1 |
| | no | % | 5.5 | 16.5 | 26.4 | 30.8 | 20.9 |
| Uber_personal_car | yes | % | 5.1 | 7.8 | 18.9 | 28.1 | 40.2 |
| | no | % | 7.7 | 13.2 | 30.8 | 34.1 | 14.3 |
| Uber_clients | yes | % | 0.6 | 1.4 | 11.4 | 31.4 | 55.1 |
| | no | % | 5.5 | 6.6 | 29.7 | 36.3 | 22.0 |
| Uber_Experience | yes | % | 0.6 | 3.2 | 10.3 | 25.2 | 60.6 |
| | no | % | 4.4 | 14.3 | 29.7 | 30.8 | 20.9 |

Source: by authors, based on research and SPSS output.

### 4.3. Uber–Taxi Conflict

One particular aspect of the analysis focused on the conflict between Uber and traditional services. Respondents were asked for their opinions on issues being debated by both sides, with five out of six referring to the elements of ridesharing regulation (Table 9):

- A total of 26% of respondents considered that Uber should definitely (5) be regulated like traditional taxi services, and the proportion was higher if we referred to the similar driver certification procedure (almost 30%);
- The payment of the same taxes as for taxis, or similar licensing, confirmed the respondents' inclination to accept that ridesharing services, although rated as better, should be subject to standardization;

■ The opinion regarding the local regulation of tariffs for Uber, as happens in the case of taxi services, was quite neutral, with the distribution of responses and the average (3.07) showing a rather indeterminate position on this issue.

**Table 9.** The mean values for variable Uber–Taxi conflict opinions.

| | Mean |
|---|---|
| Uber's business should be based on taxi services regulations and comply with its specific legislation. | 3.335 |
| Uber should pay the same taxes as taxi companies. | 3.303 |
| Uber should be licensed as a taxi. | 3.333 |
| Uber drivers should go through the same certification process as taxi drivers. | 3.537 |
| Uber tariffs should be regulated locally, similar to taxi services. | 3.07 |
| Uber is a safe service. | 4.312 |

Source: by authors, based on research.

It is worth noting that while there is general agreement on the regulation of Uber services, which is very similar to that of traditional services, the prevailing opinion on the safety of the service was clearly in favor of Uber.

Testing the H2 hypothesis (there are no significant differences between respondents who have used the Uber service and those who have not used it in terms of their attitudes towards the Uber-taxi conflict) and looking at the results of the chi-square test, the following can be observed (Table 10):

■ There were no significant differences between the opinions of respondents who used the Uber service and those who had not, in terms of opinions on the issue of the Uber–taxi conflict: ridesharing as a taxi, tax system, licensing, driver attestation, and local tariff regulations;

■ The link between the use of the Uber service and the perceptions and attitudes of respondents was negative and of low intensity in relation to the opinions on the service safety.

**Table 10.** Pearson chi-square test: Uber–taxi conflict variables.

| | UBER | |
|---|---|---|
| **VARIABLE** | **Pearson Chi-Square Value** | **Sig. (2-Tailed)** |
| **UBER_CONFLICT_LEGISLATION** | 0.042 | 0.265 |
| **UBER_CONFLICT_TAXES** | 0.048 | 0.202 |
| **UBER_CONFLICT_LICENCE** | 0.040 | 0.287 |
| **UBER_CONFLICT_DRIVERS** | 0.016 | 0.664 |
| **UBER_CONFLICT_TARRIFS** | 0.026 | 0.482 |
| **UBER_CONFLICT_SAFE** | −0.171 | 0.000 |

Source: by authors, based on research and SPSS output.

Thus, if the respondents' positions on the elements related to the regulation of the service were not influenced by the use of Uber, the position on the safety offered to customers was somewhat different: almost 60% of those who had used the service were convinced that the safety of customers was guaranteed, while the percentage was significantly lower for the other points (Table 11).

**Table 11.** Uber–taxi conflict—*"Uber is a safe service"*.

| | | | | | | | |
|---|---|---|---|---|---|---|---|
| Uber_conflict_safe | yes | % | 1.3 | 3.2 | 11.9 | 24.6 | 59.0 |
| | no | % | 2.2 | 5.5 | 27.5 | 30.8 | 34.1 |

Source: by authors, based on research and SPSS output.

### 4.4. Ethics Controversies

The controversies surrounding the business models of the collaborative economy were analyzed from the perspective of ethical principles using the example of Uber, the best-known provider on the ridesharing market. In this context, respondents were asked to agree or disagree with a series of statements designed to define their stance on the existing controversies. The results showed that

❖ Almost 65% of respondents fully agreed that Uber plays an important role in urban transport, and banning such companies is not a solution;
❖ More than half of the respondents considered that the emergence of private transport alternatives is favorable to consumers and market development;
❖ About 45% of respondents fully agreed that the quality of taxi services is extremely low, companies and taxi drivers being responsible for alternative services and that the taxi industry acts as a monopoly, strictly in its interest;
❖ The answer distribution is wider if we look to the opinion about Uber's contributions (company and drivers) to the local budget: there seemed to be no widespread belief that businesses of this type bring the same local economic benefits to the community as traditional taxi companies;
❖ The injustice and inadequacy of current legislation in view of the change in business models does not seem to convince many respondents (only half agree completely or partially with this, a third being rather neutral).

From an ethical point of view, there are the so-called neutralization techniques, which can be found in the series of statements on which the interviewees were asked for their opinion (Table 12):

✓ Denial of responsibility—trivate passenger transport legislation is unfair and not adopted to changes in business models;
✓ Appeal to higher loyalties 1—the Uber company and Uber drivers contributes to the budget by paying taxes;
✓ Denial of victims—the quality of taxi services is extremely low, and companies and taxi drivers are responsible for the success of the alternative service;
✓ Condemnation of condemners—the taxi industry is like a carter (monopoly) and companies act strictly in their interest and not in that of their customers;
✓ Appeal to higher loyalties 2—the emergence of private transport alternatives is favorable to consumers and market development;
✓ Invocation of normalcy—Uber has now an important role in urban transportation and banning it (or other similar companies) is not a solution.

**Table 12.** The mean values for variable Uber controversy opinions—neutralization techniques.

| Neutralization Techniques | Mean |
|---|---|
| Denial of responsibility | 3.623 |
| Appeal to higher loyalties 1 | 3.87 |
| Denial of victims | 4.051 |
| Condemnation of condemners | 4.03 |
| Appeal to higher loyalties 2 | 4.287 |
| Invocation of normalcy | 4.447 |

Source: by authors, based on research.

Testing the H3 hypothesis (There are no significant differences between respondents who have used the Uber service and those who have not used it in terms of their perceptions towards the controversies caused by ridesharing services) and looking at the results of the Chi Square test, the following can be observed (Table 13):

➢ There were significant differences between the opinions of respondents who used the Uber service and those who did not, in terms of opinions on controversies regarding unfairness and inadequacy of legislation, the contribution of Uber to the state budget, the benefits for competition and the important role Uber plays in urban transportation. The result agreed with the results of [39] who stated that individuals that have already used both taxi and ride sourcing tend to rate the quality-of-service performance (driver and/or vehicle fleet) provided by ride sourcing higher compared to taxis.

➢ The link between the use of the Uber service and opinions on controversies was in all cases above inverse and of low intensity.

➢ There were no significant differences between the opinions of respondents who used the Uber service and those who did not, regarding the opinion on the low quality of taxi services and the monopolistic position of taxi companies. This result was different from the one presented by [40], who stated that loyalty to a taxi service centers on transparency and safety (trust), while in VTCs it revolves around quality and comfort.

**Table 13.** Pearson chi-square test: Uber–taxi controversy variables.

| | UBER | |
| --- | --- | --- |
| **Variable** | **Pearson Chi-Square Value** | **Sig. (2-Tailed)** |
| UBER_CONTROVERSY_LAW | −0.141 | 0.000 |
| UBER_CONTROVERSY_BUDGET | −0.127 | 0.001 |
| UBER_CONTROVERSY_QUALITY | −0.091 | 0.014 |
| UBER_CONTROVERSY_TAXI | −0.085 | 0.022 |
| UBER_CONTROVERSY_COMPETITION | −0.168 | 0.000 |
| UBER_CONTROVERSY_BAN | −0.198 | 0.000 |

Source: by authors, based on research and SPSS output.

Based on the distribution of responses, it can be seen that the significant differences in 4 out of the 6 variables, which were statistically confirmed, were in the direction that users who had experience with Uber services tended to agree with the statements listed (Table 14).

**Table 14.** Uber controversy variable distribution.

| | | | **1** | **2** | **3** | **4** | **5** |
| --- | --- | --- | --- | --- | --- | --- | --- |
| Uber_controversy_law | yes | % | 3.8 | 9.7 | 29.8 | 27.9 | 28.7 |
| | no | % | 4.4 | 9.9 | 52.7 | 25.3 | 7.7 |
| Uber_controversy_budget | yes | % | 1.9 | 7.9 | 23.3 | 29.8 | 37.0 |
| | no | % | 5.5 | 8.8 | 35.2 | 29.7 | 20.9 |
| Uber_controversy_competition | yes | % | 1.3 | 2.2 | 12.1 | 30.2 | 54.3 |
| | no | % | 1.1 | 6.6 | 25.3 | 36.3 | 30.8 |
| Uber_controversy_ban | yes | % | 0.8 | 2.4 | 9.4 | 20.0 | 67.5 |
| | no | % | 3.3 | 3.3 | 27.5 | 23.1 | 42.9 |

Source: by authors, based on research and SPSS output.

From an ethical point of view, this shows that we must resort even more to neutralization techniques: denial of responsibility—the need for regulation, appeal to higher

loyalties—budget revenues, market and competitive development, invocation of normality—acceptance of change and the role played by new business models.

## 5. Conclusions

The proposed study on young Romanians aimed to draw a picture of their perceptions and attitudes towards a phenomenon that has been observed for some time: the emergence of new business models that disrupt the market and test the old paradigms. The tourism, real estate and transport markets were the first to feel the impact of the new players, who benefited from both the "sharing" trend and the opportunities offered by extensive digitalization. Airbnb and Uber have both become revolutionary companies and at the same time bring with them the controversies that accompany the changes, challenging habits and, above all, rising questions of legislation. The authors focused on the transport market (Uber), a decision conditioned by the scale of these services on the international markets and, in particular, on the Romanian market. This company has become a symbol of the sharing economy and, at the same time, of the controversies, including ethical ones, that still exist in this area.

Uber is widely used by young Romanians, with almost 90% of survey participants using the service [41]. Regarding the advantages of Uber, there is a broad consensus on the quality of the service (relationship between Uber and the customer, convenience, and speed) and the experience offered being somewhat weakened when it comes to reducing costs and the use of a personal car. Testing the statistical hypotheses shows that there are significant differences between the opinions of those who have used Uber services and those who have not.

The issue of the Uber–taxi conflict seems like a paradox: although the evaluation of Uber so far shows a positive attitude towards these new business models, there is a tendency to regulate these services in a similar way to taxis. From an ethical perspective, respondents resorted to neutralization techniques when explaining the understanding of Uber's success, justifying their position mainly by considering the emergence of these new models as normal and even beneficial for competition and consumers, rather than inappropriate legislation. The review of the hypotheses shows that there are no significant differences in terms of attitudes towards the Uber taxi conflict and opinions on the poor quality of taxi services and Uber's monopoly position. Instead, there are slightly different opinions when it comes to the safety of the service. Those who have used Uber are more convinced of this but also when it comes to understanding specific points of contention—the inappropriateness of the legislation, tax payments, consumer benefits and market development.

Analyzing the ethical controversies surrounding Uber's business model sheds light on the complex perspectives of ethical principles; similar concerns been raised by [42]. It reveals a gap in terms of the fairness and appropriateness of current legislation in relation to the changes in business models. In addition, users who have experience of the Uber service compared to those who have not used the service have identified significant differences in opinion, particularly in relation to the legislation, Uber's contribution to the national budget and the benefits to competition and urban transport. The findings highlight the use of neutralization techniques that point to the need for regulation, Uber's contribution to the national budget, and the acceptance of change and the role of new business models. In doing so, the findings emphasize the complexity of ethical evaluations related to ridesharing services and the influence of users' experiences on their perceptions.

**Managerial and business implications.** Digitalization and technology have enabled changes in business models that are now considered disruptive innovations. The collaborative economy is perhaps the most important change in behavior and consumption, or in the relationship between producers and consumers, and this is undoubtedly both an opportunity and a challenge for traditional businesses. Legislation will adapt to the new conditions and all stakeholders should finally be in favor of change, not against it, as it represents an evolution in terms of customer desires and motivations. Research shows that

young people's perceptions and attitudes are in favor of new business models that address not only basic needs but also social and environmental needs. Knowing the orientation of customers in this new context is therefore crucial for both new and traditional companies redefining themselves in line with the times.

**Research Limitations.** This study was subject to several limitations. The first relates to the representativeness of the sample—the research results cannot be generalized, the non-probabilistic sampling method did not guarantee representativeness and the sampling error could not be calculated. The second limitation relates to the lack of previous research studies on Generation Z collaborative consumption, which led to a generalized approach in the design of the questionnaire. Future research would aim at a research approach at the national level for all age categories, which would ensure representativeness through probabilistic sampling. Based on the results, a co-operative consumption model could be found. In view of the legislative changes, the research and the model could be re-evaluated after some time.

**Author Contributions:** Conceptualization, D.B. and A.L.K.; methodology, D.B.; software, G.G.; validation, G.G., A.B.C. and O.-D.C.; formal analysis, G.G.; investigation, R.-I.D.; resources, A.L.K.; data curation, G.G.; writing—original draft preparation, D.B.; writing—review and editing, D.B., G.G., A.L.K., A.B.C., R.-I.D. and O.-D.C.; visualization, O.-D.C.; supervision, R.-I.D.; project administration, D.B. and A.B.C.; funding acquisition, D.B., G.G., A.L.K., A.B.C., O.-D.C. and R.-I.D. All authors have read and agreed to the published version of the manuscript.

**Funding:** This research received no external funding.

**Institutional Review Board Statement:** Not applicable—It is worth mentioning that the authors conduct their work in accordance with the academic ethics promoted by our institutions.

**Informed Consent Statement:** Informed consent was obtained from all subjects involved in the study.

**Data Availability Statement:** Data is unavailable due to privacy or ethical restrictions.

**Conflicts of Interest:** The authors declare no conflicts of interest.

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
