# Peer review of "Youth Perspectives on Collaborative Consumption: A Study on the Attitudes and Behaviors of the Romanian Generation Z"

_sustainability, doi:10.3390/su16073028_

Round 1

Reviewer 1 Report (Previous Reviewer 1)

Comments and Suggestions for Authors

- thorough literature review, with adequate references

- I would advise against using bullet points (lines 187-195; 200-213)

- It's not very clear why each student had to fill in 10 questionnaires, 5 for female respondents and 5 for male respondents. Doesn't this approach skew your data and invalidates your results? Shouldn't each respondent fill in only 1 questionnaire, according to his real gender? Please clarify this matter. Other than that, the methodology is presented very clearly and thoroughly.

- at line 333 you state "The prevalence of Uber services among the young people included in the survey was 87.4%, significantly higher than among those who have Uber" . Could you please rephrase this in a way that makes it easier for the reader to understand your point?

- it's not clear to me how 1% of your respondents have graduated only from a general school of 8 classes (line 325) and why only 60% of your respondents stated that they are students since your survey was conducted among students of the "Business and Tourism Faculty of Bucharest University of Economic students" (line 264). 

- please clearly state if your hypothesis are either validated or invalidated, based on your results. To me, it seems that all your hypothesis are invalidated, but you should clearly state this fact yourself, rather than let the readers deduce that on their own. 

- please use the exact same wording for your hypothesis wherever you use them in text. I would suggest to use the wording from lines 246-252 since it's clear and easy to undestand unlikie lines 468-470, for example.

- besides presenting your quantitative or statistical results, you should also discuss them in terms of implications, causes and at the same time, you should compare your results with the results of other similar studies (in other countries maybe)

Comments on the Quality of English Language

I would suggest to have someone proof-read your paper. The English language in generally good, but in a few places, you using the wrong syntax, which may make your paper a bit hard to read.

One example are the lines 347, 348, 349 "The difference to those who do not use Uber is the higher income share (approx. 15%) of those with an income in the middle category (2501-3500) and the share of almost 50% of those with an income below 1500 lei". For me at least, it's very hard to understand what do you want to say with this statement. 

Author Response

Reviewer 2 Report (New Reviewer)

Comments and Suggestions for Authors

The paper is well structured but has some limitations that need to be overcome.

The main limitation can be found in the Methodology and in the Results sections, because – even if the questionnaire design takes into consideration interesting aspects as the Gen. Z “inclination” towards the use of technology and collaborative consumption, behaviors,  motivations, etc. - only the different opinions between Uber services users and non-users are at the end analyzed. In this way, in my opinion, the picture of Romanian Gen. Z people loose of complexity and the paper loose of interest. Any statistical relation that could be found between respondents’ characteristics and their opinions should improve the interest of the paper.

Introduction: it is very large and can be more focused. The described market tendencies are – in general - shareable but need to be supported by data and by the literature. How important is Uber in Romania?

Literature review: please better specify the concept of “neutralization approach” that seems to be central in the analysis and motivate the choice of this approach

Results and discussion: too many descriptive Tables and Figures that do not add a lot of information (i.e. Fig. 1 and 2). Also tables 6 and 8 (9 and 11, 12 and 14) can be merged or synthetized.

More comments in the pdf file.

Comments on the Quality of English Language

Round 2

Reviewer 1 Report (Previous Reviewer 1)

Comments and Suggestions for Authors

Thank you for your response. In my opinion, you have clearly addressed all the issues that I mentioned in my previous reviews. I have nothing more to add.

This manuscript is a resubmission of an earlier submission. The following is a list of the peer review reports and author responses from that submission.

Round 1

Reviewer 1 Report

Comments and Suggestions for Authors

- please take into account the fact that millenials are the people born between 1981 to 1996, while your study is targeting young people aged 18-26, which are part of Generation Z. Please clarify this matter. 

- please expand your introduction section. This section should clearly state your motivation for choosing this topic and provide sufficient background information for the reader to understand and evaluate your study. At the same time, at the end of this section, you should present the structure of your paper.

- the first paragraph from the literature review section is 40 lines long, which makes it for the reader to follow and understand it. I recommend splitting it into 3 or 4 paragraphs, each of them addressing a specific idea.

- in the literature review section, there should be a clear connection between the literature that you have studied and the research hypothesis which you are proposing for your study. H2 is related to the conflict betwen Uber and taxi services and H3 is related to the controversies caused by ridesharing services, but I can't find any relevant theoretical information in the literature section regarding these two problems. 

- you are stating two research questions/specific objectives, one related to the collaborative consumption behavior of young people and the other related to their attitude regarding alternative transport services, but it seems to me, that your research hypothesis are related only to the young people's attitude regarding ride sharing. Please clarify this matter.

- It's not clear what H2 ("There are no significant differences between respondents who used the Uber service and those who did not, compared to the controversies caused by the ridesharing services") is trying to prove.  You are comparing two sub-groups of respondents (those who used and those who did not use Uber) to a controversy. Maybe you were referring to their attitude regarding this controversy? Please clarify this matter

- it's not clear how you have constructed your survey. Was it done entirely by the you or did you take some items from other surveys used in the relevant literature? You are stating that the source is "by authors, based on research" but you don't mention what research (e.g. other studies) have helped you build the survey.

- at the same time, you did not state in which period was your study conducted, which is relevant to my first question (wether you are targeting millenials or gen. Z)

- please take into account that postgraduate students are students which have already completed an undergraduate degree (e.g. bachelor's).

- the results and discussion section is adequately written.

- the conclusions of your paper should discuss the three hypothesis on which your study is based. I can't find anything related to H3, regarding the controversies of this service.  

- the references are a bit out-dated. Out of 20 sources, uyou have cited only one paper published after 2020, while citing a source from 1978, one from 1985 and one from 1957.

Reviewer 2 Report

Comments and Suggestions for Authors

Dear authors,

I accepted to review your manuscript given the interesting topic that you tried to address. But, unfortunately, that remained only a single try. Your manuscript is more suitable for a blog post than a scientific communication. There is zero scientific rigor in most of your chapters. Among many issues, I am emphasizing a few.

1. The abstract and introduction do not communicate the essential positioning of your paper. Why is Romania so specific context to investigate?

2. The literature review does not exist. Neither the hypotheses development part. I am failing to see to what particular literature stream you are contributing. You are way to generic and unfocused.

3. The methodology section does not provide a full picture of what you did and why in order to collect data. 

- You did not elaborate on the variables that you were using (neither here nor in the literature review)
- In what language was your questionnaire?
- Why did you use a questionnaire?
- Did you use some overreaching theoretical framework to build up your hypotheses?
- What was your contact strategy?
- What questions did you use in your questionnaire? Did you create them? If not, where did you source them?
- The questions that you are asking seem to only vaguely correspond to your hypotheses (?).
- and many other...

Unfortunately, given the complete lack of rigor and positioning of your manuscript, I stopped reading further as it would be a waste of time.

While I understand your attempt at the domain, your manuscript is way below any publishing standard. You must invest a serious amount of time to familiarize yourself with the style and the way to write academic articles.

I am sorry that I can not be more positive, but you simply do not leave me much of a choice.